# Common limitations of performance metrics in biomedical image analysis

**Annika Reinke**[1,2,3]                                   A.REINKE@DKFZ-HEIDELBERG.DE

**Delphi consortium on metrics**[*]

**Lena Maier-Hein**[1,2,3,4]                              L.MAIER-HEIN@DKFZ-HEIDELBERG.DE

[1] *Div. Computer Assisted Medical Interventions, German Cancer Research Center (DKFZ), Germany*

[2] *HIP Helmholtz Imaging Platform, German Cancer Research Center (DKFZ), Heidelberg, Germany*

[3] *Faculty of Mathematics and Computer Science, Heidelberg University, Germany*

[4] *Medical Faculty, Heidelberg University, Germany*

## Abstract

While the importance of automatic biomedical image analysis is increasing at an enormous pace, recent meta-research revealed major flaws with respect to algorithm validation. Performance metrics are key for objective, transparent and comparative performance assessment, but little attention has been given to their pitfalls. Under the umbrella of the Helmholtz Imaging Platform (HIP), three international initiatives – the MICCAI Society's challenge working group, the Biomedical Image Analysis Challenges (BIAS) initiative, as well as the benchmarking working group of the MONAI framework – have now joined forces with the mission to generate best practice recommendations with respect to metrics in medical image analysis. Consensus building is achieved via a Delphi process, a popular tool for integrating opinions in large international consortia. The current document serves as a teaser for the results presentation and focuses on the pitfalls of the most commonly used metric in biomedical image analysis, the Dice Similarity Coefficient ($DSC$), in the categories of (1) mathematical properties/edge cases, (2) task/metric fit and (3) metric aggregation. Being compiled by a large group of experts from more than 30 institutes worldwide, we believe that our framework could be of general interest to the MIDL community and will improve the quality of biomedical image analysis algorithm validation.

**Keywords:** Segmentation, Validation, Metrics, Challenges, Good Scientific Practice.

## 1. Common limitations of segmentation metrics

Image segmentation is one of the most popular image processing tasks. An international meta-analysis showed that the chosen metrics in segmentation challenges radically influence the resulting rankings (Maier-Hein et al., 2018). Although work on clinical relevance of metrics (Vaassen et al., 2020) or data biases (Badgeley et al., 2019) exist, researchers are missing guidelines for choosing the right metric for a given problem (Maier-Hein et al., 2018). To address this community request, this document summarizes common pitfalls related to the most frequently used metric in medical image segmentation, namely the Dice Similarity Coefficient ($DSC$) (Dice, 1945). A longer version of this teaser document is available at (Reinke et al., 2021).

---

[*] **Full author list:** A. Reinke, M. Eisenmann, M.D. Tizabi, C.H. Sudre, T. Rädsch, M. Antonelli, T. Arbel, S. Bakas, M.J. Cardoso, V. Cheplygina, K. Farahani, B. Glocker, D. Heckmann-Nötzel, F. Isensee, P. Jannin, C.E. Kahn, J. Kleesiek, T. Kurc, M. Kozubek, B.A. Landman, G. Litjens, K. Maier-Hein, A.L. Martel, B. Menze, H. Müller, J. Petersen, M. Reyes, N. Rieke, B. Stieltjes, R. Summers, S.A. Tsaftaris, B.van Ginneken, A. Kopp-Schneider, P. Jäger, L. Maier-Hein.

**Full paper and affiliations:** (Reinke et al., 2021): `https://arxiv.org/abs/2104.05642`

## 1.1. Fundamental mathematical properties

Awareness of a metric's mathematical properties is crucial when determining its suitability for a given task. Segmentation of small structures, such as brain lesions, is essential for many image processing applications; however, the *DSC* may be inappropriate here (Fig. 1).

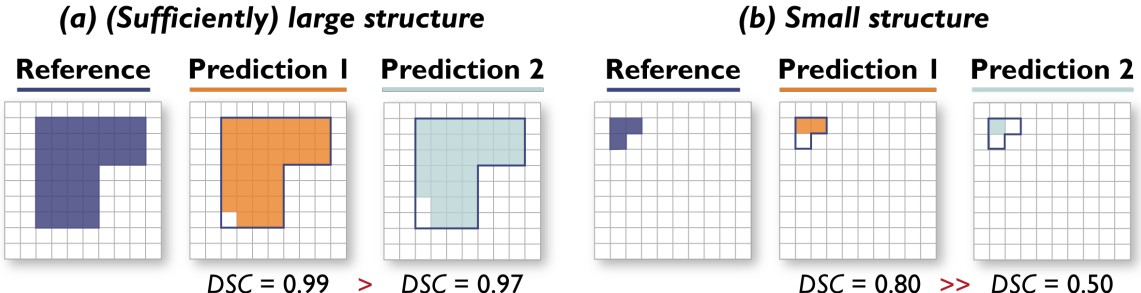

Figure 1: Effect of the **structure size** on the Dice Similarity Coefficient (*DSC*). The predictions of two algorithms (*Prediction 1/2*) differ in only a single pixel. In case of a small structure (b), this has a substantial effect on the associated metric value.

## 1.2. Suitability for underlying image processing task

While performance metrics are typically expected to reflect a domain-specific validation goal, segmentation metrics such as the *DSC* are commonly also applied to *detection and localization* tasks (Jäger, 2020). From a clinical perspective, an algorithm covering all structures of interest (e.g. tumors) would be of much higher value compared to one producing a highly accurate segmentation for one structure but missing the others. This, however, is not reflected in the *DSC* metric values, as shown in Fig. 2.

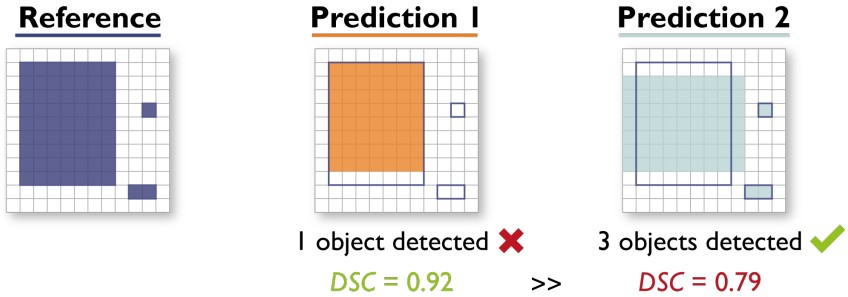

Figure 2: Effect of using a **segmentation metric for object detection**. The prediction of an algorithm only detecting one of three structures (*Prediction 1*) leads to a substantially higher *DSC* compared to that of another algorithm (*Prediction 2*) detecting all structures.

## 1.3. Metric aggregation

In international competitions, metric values are often aggregated over all test cases to produce a ranking (Maier-Hein et al., 2018). Fig. 3 illustrates the effect of missing values.

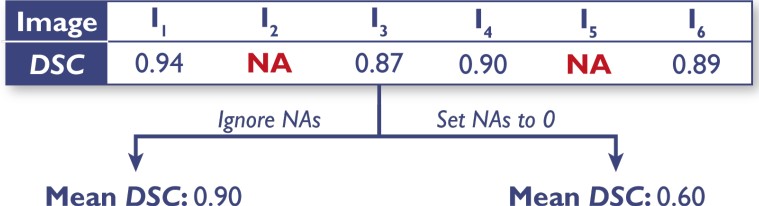

Figure 3: Effect of **missing values** when aggregating metric scores. Ignoring missing values can lead to a substantially higher *DSC* compared to setting them to the worst possible value (here: 0).

## 2. Conclusion

Choosing the right metric for a specific image processing task is non-trivial. Our MIDL presentation raises awareness about some common flaws of the most frequently used segmentation metric in the biomedical image analysis community and gives best practice recommendations for choosing the most appropriate metric(s) in an application-specific manner. Details regarding the Delphi consortium compiling the recommendations will be presented in a follow-up publication soon.

## Acknowledgments

This work was initiated by the Helmholtz Imaging Platform (HIP). It was further supported by the NIH Clinical Center Intramural Research Program, the NIH National Cancer Institute (NCI: U01CA242871) and the NIH National Institute of Neurological Disorders and Stroke (NINDS: R01NS042645).

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
