# OpenReview forum: "Common limitations of performance metrics in biomedical image analysis"
_MIDL.io/2021/Conference/Short — MIDL 2021 Poster_

### Official Review · Reviewer_XyAT · 2021-04-29

**Confidence:** 5
**Final Rating:** 3

**Summary:**

The paper introduce an initiative to establish best practice recommendations for validation metrics in biomedical imaging. The need for such an initiative is demonstrated by highlighting limitations of the commonly used Dice coefficient in the context of segmentation tasks. An important aspect of this is the suggestion to focus the analysis on three categories , mathematical properties, task/metric fit and metric aggregation.

**Strengths:**

The idea of an agreed best practice for validation metrics is good and something we need.

I like the idea of very clearly demonstrating when a metric is suitable and when it is not. Both Figure 1 and 2 does this well.

The example with Dice nicely illustrates the structure for how to analyze a metric according to the three categories.

**Weaknesses:**

In my view, the most important point in the paper is the Delphi consortium on metrics, which is only mentioned in the abstract. I would much rather have read a paper focusing on the consortium, how it will work, the guiding principles, the goals and how I can influence the recommendations.

Although toy examples are great for illustrating the point, these points need to be demonstrated in practice. I think the highlighted  limitations of Dice are well known to many in the field, yet has not stopped it from becoming metric of choice in many cases. A couple of cases showing when it has lead to flawed conclusions would be more powerful than the small illustrations.

The analysis of Dice is a bit to superficial to be really interesting. I am aware that space is limited, but I would have liked to see some ideas for how to perform analysis according to the three categories. For example, in 1.1 it would be interesting to see how Dice behaves as we change from large to small structure and try to define what a small structure is when we use Dice.

**Deanonymize Review:**

no

**Detailed Comments:**

It might be relevant to consider the framework for evaluating metrics in the technical report from Powers focusing on how measures are biased wrt imbalance in data and predictions.

David M W Powers
Evaluation: From Precision, Recall and F-Factor to ROC, Informedness, Markedness & Correlation
School of Informatics and Engineering Flinders University • Adelaide • Australia
http://david.wardpowers.info/BM/Evaluation_SIETR.pdf

**Justification Of The Rating:**

As mentioned above, I would have liked to see more on how we get to a recommendation on best practice. Alternatively I would liked a bit more substance in the analysis. As is, the paper falls between "presenting the initiative" and "showing that Dice can be problematic".


**Paper Type:**

validation/application paper

**Special Issue:**

no

---

### Official Review · Reviewer_R3r1 · 2021-04-30

**Confidence:** 4
**Final Rating:** 3

**Summary:**

In this paper a consortium highlights the need to generate best practice recommendations with respect to metrics in medical image analysis. They show this in the example of the Dice Similarity Coefficient (DSC), in the categories of (1) mathematical properties/edge cases, (2) task/metric fit and (3) metric aggregation.

**Strengths:**

Initiatives like these are really important and necessary followups for the greater medical imaging community after the neuroimaging community has taken steps in this direction years ago in the OHBM Committee on Best Practices in Data Analysis and Sharing (COBIDAS) report (http://www.humanbrainmapping.org/files/2016/COBIDASreport.pdf). All points regarding the mathematical properties, the suitability for the task and aggregation of scores are sensibly described.

**Weaknesses:**

The suitability for publishing this teaser at a conference is questionable. Why does the consortium not publish a full report, just as the OHBM COBIDAS did insteadf of presenting a teaser at a conference? The pitfalls of the DSC described are all accurate, but they are also known and hence there's no novelty.

**Deanonymize Review:**

no

**Justification Of The Rating:**

Initiatives to standarize data and analysis practices are very important, on the other hand I am a bit doubtful about the suitability of a conference to publish such a short teaser instead of publishing a full report by the consortium itself.

**Paper Type:**

methodological development

**Special Issue:**

no

---

### Meta-Review · Area_Chair_KRgm · 2021-05-06

**Recommendation:** Accept (Poster)
**Confidence:** 5

**Metareview:**

The reviewers raise some valid questions about the positioning of the paper, but generally agree that it is worth presenting at MIDL.

---

### Decision · Program_Chairs · 2021-05-11

Accept (Poster)